# Integrating field surveys and remote sensing to optimize phosphorus resource management for rainfed rice production in the Central plateau of Burkina Faso

**Shinya Iwasaki**[1]*, **Takashi Kanda**[1], **Satoshi Nakamura**[1], **Satoshi Uchida**[1], **Simporé Saïdou**[2], **Albert Barro**[2], **Fujio Nagumo**[1]

**1** Japan International Research Center for Agricultural Sciences, Tsukuba, Ibaraki, Japan, **2** Institut de l'Environnement et de Recherche Agricole, Ouagadougou, Burkina Faso

* iwasakis@affrc.go.jp

## Abstract

Rice production in sub-Saharan Africa (SSA) is restricted by low water availability, soil fertility, and fertilizer input, and phosphate rock (PR) application is expected to increase production. Soil water conditions and soil types affect the efficacy of phosphorus fertilization in improving productivity. However, these factors are rarely discussed together. In this study, we aimed to investigate the soil types and soil water conditions in the fields, as well as their effects on rice productivity after phosphorus fertilization, and optimize the findings using remote sensing techniques. A soil profiling survey, followed by a field experiment in seven farmer fields, was performed in the Central plateau of Burkina Faso. The following treatments were applied: nitrogen and potassium fertilization without phosphorus (NK), PR application with NK (NK+PR), and triple super phosphate (TSP) application with NK (NK+TSP). Submergence duration and cumulative water depth were recorded manually. The inundation score, estimated using a digital elevation model, explained the distribution of soil types and soil water conditions and correlated negatively with sand content and positively with silt and clay content, indicating an illuvial accumulation of fine soil particles with nutrient transportation. The field experiment showed that although grain yield was significantly restricted by phosphorus deficiency, the increase in yield after phosphorus fertilization was higher in Lixisols and Luvisols than in Cambisols because of the low Bray-2-phosphorus content of Lixisols and Luvisols. The inundation score correlated positively with grain yields after NK+PR and NK+TSP treatments. In conclusion, soils with low inundation scores (mainly Lixisols and Luvisols) showed a drastic increase in grain yield after TSP application, whereas those with high inundation scores showed comparable yields after PR and TSP application despite the low phosphorus fertilization effect. Our findings would help optimize fertilization practices to increase rice productivity in SSA.

**Data Availability Statement:** All relevant data are within the manuscript and its Supporting Information files.

**Funding:** This work was financially supported by the Science and Technology Research Partnership for Sustainable Development (SATREPS) project No. JPMJSA1609, Japan Science and Technology Agency (JST), and Japan International Cooperation Agency (JICA) (Project on the establishment of the model for fertilizing cultivation promotion using Burkina Faso phosphate rock, No. JPMJSA1609). There was no additional external funding received for this study. The funders had no role in study design, data collection and analysis, decision to publish, or preparation of the manuscript.

**Competing interests:** The authors have declared that no competing interests exist.

## Introduction

Rice demand and production have increased in Sub-Saharan Africa (SSA) over the years. From 1991–2001, rice cultivation accounted for only 1.76% of agricultural production in the region; however, this figure rose to 3.96% from 2002–2013 [1]. Despite the increase in cultivation, the average grain yields in SSA were reported to be 4.0 Mg ha$^{-1}$ for irrigated lowlands, 2.6 Mg ha$^{-1}$ for rainfed lowlands, and 1.6 Mg ha$^{-1}$ for rainfed uplands, significantly lower than the potential rice yield that could be obtained from the region [2]. Limited water availability, soil fertility, and fertilizer input are the factors that hinder rice production in the region [3, 4]. Rainfed agricultural production in SSA is highly sensitive to spatial and temporal variability in precipitation [5]. Additionally, highly weathered soils with low phosphorus (P) contents are prevalent in SSA and severely limit rice production [6]. Although the application of P fertilizers is the most common approach for overcoming soil P deficiency [7], its adoption by local farmers is limited because of the high cost of imported fertilizers [8] and high variability in their fertilization efficacy. A potential alternative source of P fertilizers is the widespread deposits of phosphate rocks (PRs) throughout SSA [9]. For example, Burkina Faso alone has 100 million metric tons of low-grade PRs that can serve as an alternative P fertilizer source [10, 11]. However, the efficacy of P fertilizers is impacted by several factors, such as P solubility of fertilizer, mean submergence duration and groundwater level (GWL) [12–15].

Soil type is another critical factor impacting crop production and fertilization efficacy in SSA [16]. A study in SSA has investigated the relationship between the soil type and rainfed lowland rice productivity [17]. In lowland soils, the distribution of soil types and soil water conditions is closely related, as both these factors are affected by the topography [18–21].

The Central plateau of Burkina Faso is a flat region situated in the upper areas of the Black and White Volta Rivers, with extremely gradual slopes. The river bottom zone of this region is characterized by seasonal or occasional flooding, largely determined by topographical and environmental factors. The flooding pattern in SSA has been estimated using deterministic, hydrodynamic, and probabilistic models [22–25]. However, as these models cannot be applied directly to assess flooding levels in the Central plateau of Burkina Faso because of the lack of hydrometric data, a remote sensing approach has been suggested to be relatively more practical for evaluating flooding tendencies in this region [26]. Although maps have been created to indicate the appropriate type and amount of fertilizer application in Africa in recent years [27, 28], lowland rice-producing areas are yet to be included in these maps. Specifically, the factors that impact P fertilization efficacy in lowland rice-producing areas remain to be investigated.

Therefore, in this study, we aimed to evaluate the soil types and soil water conditions in rice cultivation areas in Burkina Faso and analyzed their useability in recommending P fertilizers for rice production. Herein, we developed a new strategy for assessing flooding levels in the Central plateau of Burkina Faso using a digital elevation model (DEM) in combination with the geographic information system (GIS) to overcome the limitation of the lack of hydrometric data for this region. Our findings would contribute to the development of fertilization practices to help increase rice productivity in the upper reaches of rivers in Burkina Faso and SSA.

## Materials and methods

### Study site

This study was conducted in the Central plateau of Burkina Faso, specifically in the seven rainfed lowland rice cultivation fields (Nassoulou, Siguinvouse, Ramongo, Nandiala, Villy, Poa, and Sissene, named after the name of the village wherein they are located) around the Institute of Environment and Agricultural Research (INERA) Saria Station (12˚16´ N, 2˚0´ W)

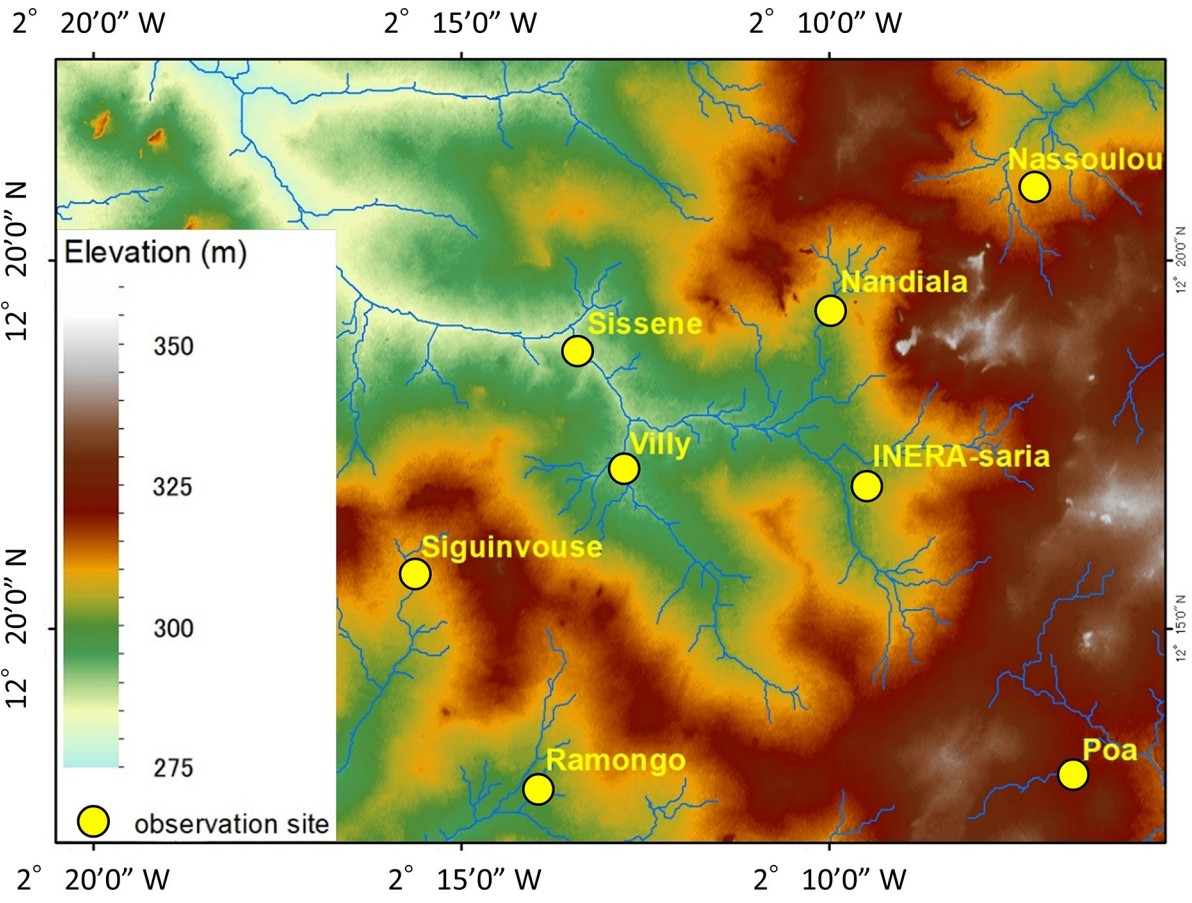

**Fig 1. Observation points overlaid onto the topographical map of the study area.**

(Fig 1). These seven fields were selected to cover the entire watershed region upstream to downstream. Permission for the survey was obtained by INERA from the farmers who owned these seven rice cultivation fields. The study area receives a monomodal rainfall pattern between May and October, with the mean annual precipitation and temperature values of 800 mm and 28°C, respectively, as recorded at the INERA Saria station [14]. The mean annual potential evaporation in the study area is estimated to be between 1,700 and 2,000 mm [29].

## Soil profiling survey and analysis

A soil profiling survey was conducted between October and November 2018 and March 2019. Composite soil samples were collected from each horizon to determine their physicochemical properties. The soil profiles were described and classified following the World Reference Base for Soil Resources system [30].

Soil particle size distribution was determined by degrading soil organic matter with $H_2O_2$ and dispersing it with sodium hexametaphosphate. Coarse and fine sand fractions (0.2–2 mm and 0.02–0.2 mm, respectively) were measured by sieving, whereas silt (0.002–0.02 mm) and clay (<0.002 mm) fractions were measured using the pipette method. Soil texture class was determined based on the textual triangle (30).

Total carbon (C) and nitrogen (N) contents were determined using the dry combustion method using an NC Analyzer (NC 220F; Sumika Chemical Analysis Service, Tokyo, Japan).

The pH and electrical conductivity of the 1:5 soil: solution suspensions in deionized water and in 1M potassium chloride (KCl) were measured using a pH electrode (LAQUA F-72; HORIBA Ltd., Kyoto, Japan) and conductivity meter (ES-51; HORIBA Ltd., Japan), respectively.

Available P was determined using the Bray1 and Bray2 methods [31]. Phosphorous concentrations were determined calorimetrically using a spectrophotometer (UV-1800; Shimadzu Corp., Osaka, Japan). Exchangeable cations were extracted using 1.0 M ammonium acetate (pH 7.0), and the cation concentrations were determined using an inductively coupled plasma atomic emission spectrophotometer (ICPE-9000; Shimadzu Corp., Osaka, Japan).

After the exchangeable base extraction, the residues were washed with 80% (v/v) ethanol to determine their cation exchange capacity (CEC). Saturated $NH_4^+$ ions were extracted 4 times with 1 M KCl (pH 7.0), and the concentrations were determined using the salicylate method [32] and continuous-flow analysis in an Auto Analyzer III (BL-TEC KK, Tokyo, Japan).

The base saturation (BS) was calculated as the ratio of exchangeable basic cations to the CEC. The effective BS (EBS) [30] was calculated as the sum of exchangeable base cations ($Ca^{2+}+Mg^{2+}+K^++Na^+$) divided by the effective CEC ($Ca^{2+}+Mg^{2+}+K^++Na^++Al^{3+}$).

## Field experiment

We conducted a field experiment to assess the efficacy of directly applying PR in rice production. The experiment was performed over 3 years, starting in 2013. The experimental plots were managed following local farmer practices, with rainfed cultivation utilizing floodwater, and no puddling or leveling was performed. Each plot was 25 $m^2$ in size and was enclosed in a 30-cm-high bund to prevent fertilizer overflow and collect water. The following four treatments were used in this study:

1. No fertilization (CT)

2. Application of N and potassium (K) without P fertilization (NK)

3. Application of PR with NK (NK+PR)

4. Application of triple superphosphate (TSP) with NK (NK+TSP)

The application rates for N, P, and K were 90 kg $ha^{-1}$, 60 kg $ha^{-1}$, and 50 kg $ha^{-1}$, respectively. The source of P used was PR from the Kodjari deposit (12˚1′ N; 1˚55′ E) in Burkina Faso, which contained 113 g P $kg^{-1}$. The solubility of P in the PR for water, alkaline ammonium citrate, and 2% citric acid were 0.2, 2.5, and 31.1%, respectively. Moreover, N and K were applied as ammonium sulfate and KCl, respectively. In 2015, P was not applied. Rice seeds were directly sown at a spacing of 20 cm × 20 cm at the beginning of July 2015, and the plants were harvested in mid-October 2015. The duration of submergence and water depth were measured manually.

## Inundation score estimation using a DEM

The flow of inundation score estimation and the following analysis are shown in Fig 2. In this study, the ALOS World 3D-30m DEM with 5 m resolution was used (https://www.eorc.jaxa.jp/ALOS/en/aw3d30/data/index.htm). The dataset provides a digital surface model, including anthropogenic construction and large trees that may show higher elevation values than ground-level ones. However, as the ground surface in this region is usually covered with bare soil or low vegetation during the dry season, the difference between the DEM and digital surface model was assumed to be negligible.

The inundation score was estimated based on water accumulation scores and convexities. First, the water accumulation score was calculated using ArcGIS® v. 10.7.1 software to

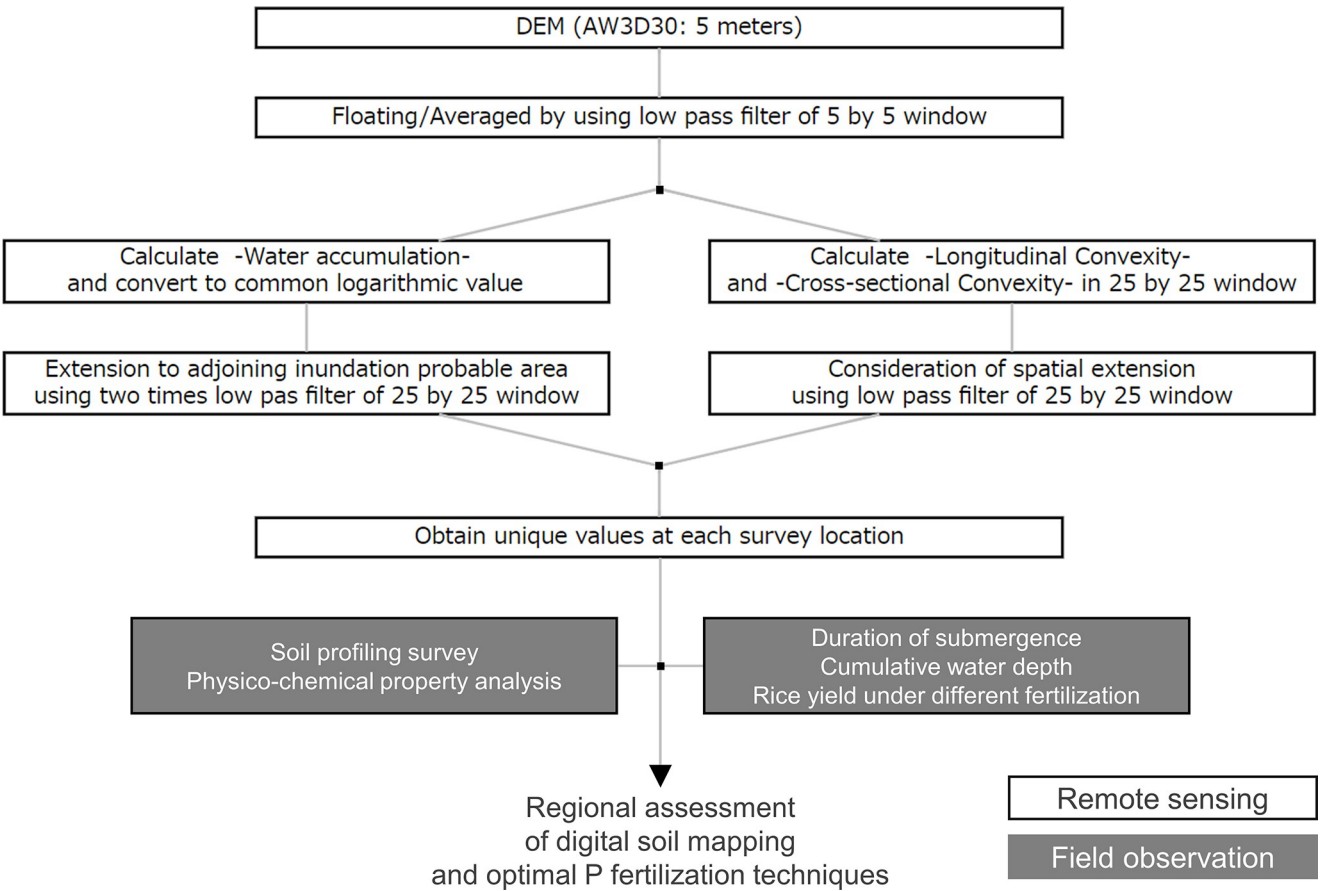

**Fig 2. Flowchart of the inundation score estimation process and the study design.** DEM, digital elevation model; AW3D30, ALOS World 3D-30m DEM (https://www.eorc.jaxa.jp/ALOS/en/aw3d30/data/index.htm); P. phosphorus.

represent the accumulated number of cells on the upstream side and the amount of water from rainfall. The DEM data in geographical coordinates were converted to the Universal Transverse Mercator system. Subsequently, a low-pass filter with a window size of 5 × 5 was used to convert the integer values to floating values. Logarithmic transformation was applied to the water accumulation score, and a low-pass filter with a window size of 25 × 25 was applied twice. As a 25 × 25 spatial filter was used to eliminate the insignificant effect of microtopography, the longitudinal and cross-sectional convexities were estimated using ENVI® v. 5.5.1. A negative convexity indicates concave topography, inducing water stagnation. Finally, the inundation score was calculated by multiplying the water accumulation score by convexities. In this calculation, scores are shown as absolute values, and when convexity was positive, water was assumed not to stagnate, and the inundation score was treated as zero.

## Statistical analyses

The R version 4.2.1 software [33] was used to process and visualize the data. Considering that the data was not normally distributed, Spearman's correlation coefficient was used to demonstrate the relationships between soil physicochemical properties. A one-way analysis of variance followed by Tukey's honestly significant difference multiple comparisons test was utilized to analyze the differences in grain yield between the fertilization treatments for three soil groups (Lixisols and Luvisols, Cambisols, and all sites separately). The differences in the grain

yield of the same treatment across different soil types were analyzed using non-paired $t$-tests. Principal component analysis (PCA) was applied to understand the variability of the data and identify the characteristics of each site and the soil type by reducing the dimension of the data while minimizing the loss of information [34, 35]. After the standardization, cumulative precipitation, water accumulation score, longitudinal and cross-sectional convexity, submergence duration, cumulative water depth, and soil particle distribution data were subjected to PCA. Single regression models were applied to investigate the relationship between inundation scores and soil particle distribution, submergence duration, cumulative water depth, and grain yield in each treatment. The possibility of substituting TSP with PR was demonstrated by illustrating the relationship between the inundation scores and the relative grain yields obtained after the NK+PR and NK +TSP treatments. An alpha level of significance of 0.05 was used throughout the study.

## Results

### Soil classification and physicochemical properties

Table 1 describes the physicochemical properties of the three soil groups. Supplementary materials (S1 Table and S1 Fig) show soil morphological properties and photographs of the soil profiles. The seven soil profiles exhibited gleyic conditions from the middle to lower horizons; however, reducing conditions were not detected using dipyridyl solution testing. Therefore, no soil was classified as a Gleysol. The soil types were classified based on the presence of argic horizons, CEC per clay in argic horizons, and ferric horizons. Soils at Nassoulou, Siguinvouse, Ramongo, and Poa sites had argic horizons starting ≤100 cm below the soil surface. The CEC per clay in the argic horizons of Siguinvouse, Ramongo, and Poa was <24 $cmol_c$ $kg^{-1}$ clay, whereas that in the argic horizon of Nassoulou was >24 $cmol_c$ $kg^{-1}$ clay. The EBS value was >50% for all horizons between 50 and 100 cm. Notably, Bmvg (46–73 cm) and Bmvg2 (73–100 cm) at Ramongo and Btwgc (42–55 cm) and Bmv (55–90 cm) at Siguinvouse were ferric horizons (detailed definitions of the horizons are provided in [30]). Thus, the soils at Ramongo and Siguinvouse were classified as Ferric Relictigleyic Lixisols, whereas the soil at Poa was classified as Abruptic Relictigleyic Lixisol. The soil at Nassoulou was classified as Abruptic Relictigleyic Luvisol. Cambic horizons (Bwg1 and Bwg2 in Sissene, Bwg1 in Nandiala, and Bwg1 in Villy) were present in the soils at Sissene, Nandiala, and Villy, as confirmed by their distinct color and clay content compared with that of their underlying horizons. Additionally, EBS was ≥50% throughout the soil at these sites. Sideralic Bwg1 (21–31 cm), Bwg2 (31–40 cm) at Sissene, and Bwg1 (19–35 cm) at Villy were present, as their CEC per clay was <24 $cmol_c$·$kg^{-1}$ clay. Therefore, the soils were classified as Eutric Sideralic Relictigleyic Cambisol at Sissene and Villy and Eutric Relictigleyic Cambisol at Nandiala. The mean elevation of the Lixisols and Luvisols group and Cambisols group was 312.8 and 296.7 m, respectively. The surface horizon of the Cambisols group had a finer texture with higher silt and clay contents than those in the group containing Lixisols and Luvisols (Table 1). Total C and N content, CEC per clay were higher for the Cambisols than for the Luvisols and Lixisols in the entire soil profiles. Sand content correlated negatively with total C and N content, exchangeable cations, and CEC (Table 2). In contrast, silt content showed a positive correlation with total C and N content and Bray2-P. Clay content correlated positively with the concentrations of exchangeable cations and CEC.

### Soil water conditions and rice grain yields under different fertilization treatments

Soil water conditions and grain yields and a comparison of these variables across the different soil types are presented in Table 3. The results indicate that Cambisols had longer

**Table 1. Physicochemical properties of soil from the seven soil profiles in the rice cultivation area.**

| Site and soil type | Horizon | Depth | Texture | Sand | Silt | Clay | Total C | Total N | pH (H₂O) | pH (KCl) | EC | Bray1-P | Bray2-P | Al | Exchangeable cations | | | | CEC | BS | EBS |
|---|---|---|---|---|---|---|---|---|---|---|---|---|---|---|---|---|---|---|---|---|---|
| | | cm | | % | % | % | $g\,kg^{-1}$ | $g\,N\,kg^{-1}$ | | | $S\,m^{-1}$ | $mg\,P\,kg^{-1}$ | $mg\,P\,kg^{-1}$ | $cmol_c\,kg^{-1}$ | Na $cmol_c\,kg^{-1}$ | K $cmol_c\,kg^{-1}$ | Ca $cmol_c\,kg^{-1}$ | Mg $cmol_c\,kg^{-1}$ | $cmol_c\,kg^{-1}$ | % | % |
| Nassoulou — Abruptic Relictigleyic Luvisols (LV-rl.ap) | Apg | 0–6 | SL | 77.0 | 11.1 | 11.9 | 4.4 | 0.39 | 5.1 | 4.1 | 5.70 | 1.8 | 3.3 | 0.42 | 0.04 | 0.12 | 1.42 | 0.57 | 3.8 | 56.2 | 83.7 |
| | ABtg | 6–24 | LiC | 50.8 | 14.3 | 34.9 | 4.9 | 0.48 | 6.0 | 4.6 | 2.18 | 0.6 | 2.0 | 0.08 | 0.09 | 0.17 | 5.29 | 2.05 | 10.1 | 75.0 | 99.0 |
| | Btwgc1 | 24–35 | LiC | 43.2 | 12.9 | 43.9 | 4.1 | 0.49 | 6.2 | 4.7 | 1.88 | 0.3 | 1.0 | 0.07 | 0.09 | 0.18 | 5.67 | 2.11 | 11.2 | 72.0 | 99.2 |
| | Btwgc2 | 35–50 | HC | 39.2 | 11.5 | 49.3 | 2.8 | 0.36 | 6.4 | 5.0 | 2.16 | 0.3 | 1.8 | 0.07 | 0.12 | 0.28 | 6.43 | 2.70 | 17.6 | 54.2 | 99.3 |
| | Btwgc3 | 50–70 | HC | 39.5 | 10.5 | 50.0 | 2.4 | 0.31 | 6.5 | 5.2 | 2.40 | 0.4 | 1.8 | 0.07 | 0.12 | 0.34 | 6.90 | 3.05 | 16.3 | 64.0 | 99.4 |
| | Bmv | 70–80+ | | 53.9 | 6.8 | 39.3 | | | | | | | | | | | | | | | |
| Siguinvouse — Ferric Relictigleyic Lixisols (LX-rl.fr) | Apg | 0–5 | SL | 74.8 | 14.1 | 11.1 | 6.2 | 0.62 | 4.6 | 4.1 | 10.96 | 1.6 | 3.5 | 0.29 | 0.07 | 0.07 | 0.70 | 0.12 | 2.4 | 40.9 | 76.7 |
| | ABwg1 | 5–15 | SL | 72.3 | 13.4 | 14.3 | 2.5 | 0.27 | 5.2 | 3.9 | 1.50 | 2.5 | 2.0 | 0.83 | 0.04 | 0.03 | 0.26 | 0.02 | 2.4 | 13.9 | 28.9 |
| | ABwg2 | 15–27 | SCL | 65.9 | 15.3 | 18.8 | 1.9 | 0.23 | 5.4 | 3.9 | 1.14 | 1.5 | 1.5 | 0.74 | 0.03 | 0.04 | 0.66 | 0.12 | 3.2 | 26.1 | 53.3 |
| | Btwg | 27–42 | LiC | 53.9 | 13.7 | 32.4 | 2.0 | 0.29 | 5.6 | 4.0 | 1.27 | 0.5 | 1.3 | 0.53 | 0.04 | 0.08 | 1.55 | 0.40 | 6.8 | 30.6 | 79.5 |
| | Btwgc | 42–55 | HC | 43.6 | 6.9 | 49.5 | 2.0 | 0.31 | 5.7 | 4.1 | 1.27 | 0.2 | 1.3 | 0.50 | 0.07 | 0.20 | 2.99 | 0.91 | 10.6 | 39.2 | 89.3 |
| | Bmv | 55–90+ | LiC | 53.9 | 6.8 | 38.6 | | | | | | | | | | | | | | | |
| Ramongo — Ferric Albic Relictigleyic Lixisols (LX-rl.fr) | Apg1 | 0–5 | SL | 82.3 | 10.8 | 6.9 | 6.7 | 0.58 | 6.0 | 5.3 | 3.53 | 6.1 | 9.8 | 0.03 | 0.04 | 0.13 | 1.74 | 0.44 | 2.3 | 100.4 | 98.9 |
| | Apg2 | 5–17 | SL | 78.6 | 11.5 | 9.8 | 3.4 | 0.31 | 6.0 | 4.2 | 1.62 | 1.2 | 1.5 | 0.17 | 0.02 | 0.10 | 0.66 | 0.14 | 1.9 | 47.3 | 84.6 |
| | Btgw1 | 17–30 | SCL | 74.4 | 9.1 | 16.5 | 2.0 | 0.24 | 5.6 | 4.1 | 1.25 | 1.5 | 1.3 | 0.40 | 0.02 | 0.11 | 0.59 | 0.15 | 2.4 | 35.8 | 68.3 |
| | Btgw2 | 30–46 | SC | 55.8 | 12.4 | 31.9 | 2.2 | 0.29 | 5.3 | 4.0 | 1.54 | 0.6 | 2.0 | 0.69 | 0.02 | 0.13 | 1.12 | 0.31 | 4.8 | 32.7 | 69.5 |
| | Bmvg | 46–73 | LiC | 52.4 | 9.0 | 48.3 | | | | | | | | | | | | | | | |
| | Bmvg2 | 73–100+ | LiC | 49.9 | 9.1 | 41.0 | | | | | | | | | | | | | | | |
| Nandiala — Eutric Relictigleyic Cambisols (CM-rl.eu) | Apg | 0–21 | LiC | 37.0 | 20.6 | 42.3 | 16.0 | 1.17 | 5.7 | 3.9 | 2.33 | 0.9 | 9.5 | 1.30 | 0.09 | 0.34 | 5.14 | 1.99 | 15.0 | 50.6 | 85.3 |
| | Bwg1 | 21–45 | LiC | 51.7 | 13.8 | 34.5 | 7.0 | 0.49 | 6.1 | 4.8 | 1.55 | 1.0 | 3.0 | 0.07 | 0.06 | 0.17 | 5.50 | 2.10 | 11.6 | 67.4 | 99.1 |
| | Bwg2 | 35–65 | SC | 56.1 | 9.9 | 34.0 | 3.1 | 0.34 | 6.4 | 5.1 | 1.48 | 0.5 | 2.3 | 0.03 | 0.04 | 0.16 | 3.63 | 1.58 | 8.9 | 60.8 | 99.4 |
| | Bwg3 | 65– | SC | 57.9 | 9.2 | 32.9 | 2.4 | 0.31 | 6.5 | 5.1 | 1.58 | 0.5 | 2.5 | 0.04 | 0.04 | 0.18 | 3.29 | 1.49 | 9.4 | 53.4 | 99.2 |
| Villy — Eutric Sideralic Relictigleyic Cambisols (CM-rl.se.eu) | Apg | 0–19 | HC | 21.4 | 30.3 | 48.3 | 9.1 | 0.82 | 5.2 | 3.7 | 2.46 | 1.9 | 10.3 | 1.33 | 0.12 | 0.22 | 2.96 | 0.89 | 11.2 | 37.6 | 75.9 |
| | Bwg1 | 19–35 | HC | 9.6 | 24.3 | 66.1 | 9.1 | 0.82 | 5.6 | 3.9 | 2.18 | 3.9 | 11.5 | 1.08 | 0.21 | 0.34 | 4.87 | 1.62 | 14.2 | 49.7 | 86.7 |
| | Bwg2 | 45–75 | HC | 19.0 | 28.6 | 52.3 | 7.1 | 0.58 | 5.7 | 4.0 | 2.20 | 0.8 | 3.3 | 0.37 | 0.20 | 0.28 | 4.93 | 1.60 | 13.2 | 53.1 | 95.0 |
| | Bwg3 | 75– | HC | 10.8 | 17.6 | 71.6 | 5.2 | 0.51 | 5.7 | 3.8 | 1.86 | 0.6 | 2.8 | 1.25 | 0.24 | 0.27 | 4.80 | 1.46 | 14.9 | 45.6 | 84.4 |
| Poa — Abruptic Relictigleyic Lixisols | Apg | 0–9 | CL | 55.0 | 25.4 | 19.6 | 9.7 | 0.8 | 5.6 | 4.3 | 3.86 | 1.9 | 4.8 | 0.14 | 0.04 | 0.08 | 2.43 | 0.47 | 5.5 | 55.1 | 95.6 |
| | ABtwg | 9–28 | LiC | 43.5 | 16.3 | 40.2 | 4.3 | 0.39 | 5.4 | 3.8 | 1.21 | 1.0 | 2.0 | 1.09 | 0.03 | 0.04 | 1.70 | 0.34 | 8.0 | 26.4 | 66.0 |
| | Btwg | 28–47 | HC | 22.3 | 11.4 | 66.3 | 5.2 | 0.5 | 5.2 | 3.7 | 1.36 | 0.9 | 2.0 | 2.91 | 0.05 | 0.08 | 2.24 | 0.46 | 11.7 | 24.1 | 49.4 |

(Continued)

**Table 1.** (Continued)

| Site and soil type | Horizon | Depth cm | Texture | Sand % | Silt % | Clay % | Total C g kg⁻¹ | Total N g N kg⁻¹ | pH (H₂O) | pH (KCl) | EC S m⁻¹ | Bray1-P mg P kg⁻¹ | Bray2-P mg P kg⁻¹ | Exchangeable cations Al cmol_c kg⁻¹ | Na cmol_c kg⁻¹ | K cmol_c kg⁻¹ | Ca cmol_c kg⁻¹ | Mg cmol_c kg⁻¹ | CEC cmol_c kg⁻¹ | BS % | EBS % |
|---|---|---|---|---|---|---|---|---|---|---|---|---|---|---|---|---|---|---|---|---|---|
| (LX-rl.ap) | Btwgc1 | 47–60 | LiC | 47.3 | 9.0 | 43.7 | 2.7 | 0.33 | 5.1 | 3.8 | 1.23 | 0.9 | 2.3 | 1.80 | 0.02 | 0.08 | 1.15 | 0.28 | 7.8 | 19.7 | 46.2 |
|  | Btwgc2 | 60–100+ | LiC | 41.6 | 27.6 | 30.8 | 1.7 | 0.24 | 5.2 | 3.8 | 1.08 | 0.7 | 1.8 | 1.32 | 0.02 | 0.10 | 1.07 | 0.36 | 7.5 | 20.6 | 54.0 |
| Sissene | Apg1 | 0–3 | LiC | 39.6 | 29.9 | 30.5 | 11.9 | 0.94 | 6.2 | 4.3 | 2.21 | 1.9 | 7.0 | 0.18 | 0.06 | 0.30 | 3.46 | 1.22 | 8.4 | 60.4 | 96.6 |
| Eutric Sideralic Relictigleyic | Apg2 | 3–11 | LiC | 44.1 | 28.1 | 27.8 | 8.7 | 0.67 | 6.0 | 4.2 | 1.54 | 1.3 | 4.3 | 0.23 | 0.04 | 0.17 | 3.23 | 1.11 | 7.7 | 59.0 | 95.1 |
| Cambisols | ABwg | 11–21 | LiC | 46.8 | 25.4 | 27.8 | 6.4 | 0.48 | 5.6 | 4.0 | 1.38 | 0.8 | 1.8 | 0.78 | 0.03 | 0.14 | 1.96 | 0.65 | 6.3 | 44.1 | 78.0 |
| (CM-rl.se.eu) | Bwg1 | 21–31 | CL | 54.5 | 21.5 | 24.0 | 3.4 | 0.36 | 5.6 | 4.1 | 1.36 | 0.5 | 1.3 | 0.48 | 0.02 | 0.13 | 1.47 | 0.51 | 4.3 | 49.1 | 81.5 |
|  | Bwg2 | 31–41 | SC | 56.5 | 17.7 | 25.7 | 2.5 | 0.33 | 5.8 | 4.3 | 1.36 | 0.7 | 1.5 | 0.21 | 0.03 | 0.19 | 1.65 | 0.72 | 6.0 | 42.8 | 92.6 |
|  | Bwgc1 | 41–71 | SCL | 60.7 | 14.5 | 24.8 | 1.9 | 0.28 | 5.8 | 4.4 | 1.43 | 0.5 | 1.8 | 0.10 | 0.03 | 0.21 | 1.52 | 1.12 | 7.0 | 41.2 | 96.5 |
|  | Bwgc2 | 71– | SC | 57.7 | 10.5 | 31.7 | 1.5 | 0.26 | 6.1 | 4.8 | 1.44 | 0.2 | 1.0 | 0.03 | 0.05 | 0.30 | 1.82 | 2.01 | 7.9 | 52.7 | 99.4 |

C, carbon; N, nitrogen; EC, electrical conductivity; BS: base saturation; EBS: effective base saturation; CEC: cation exchange capacity. SL, sandy loam; LiC, light clay; HC, heavy clay; SCL, sandy clay loam; SC, sandy clay.

**Table 2. Relationships between soil particles and physicochemical properties (n = 34).**

| | Unit | Sand | | Silt | | Clay | |
|---|---|---|---|---|---|---|---|
| **Sand** | % | | | | | | |
| **Silt** | | −0.48 | ** | | | | |
| **Clay** | | −0.93 | ** | 0.12 | | | |
| **Total C** | g kg$^{-1}$ | −0.37 | * | 0.62 | ** | 0.15 | |
| **Total N** | g kg$^{-1}$ | −0.42 | * | 0.61 | ** | 0.22 | |
| **pH (H$_2$O)** | | −0.06 | | −0.14 | | 0.13 | |
| **pH (KCl)** | | 0.31 | | −0.39 | * | −0.18 | |
| **EC** | S m$^{-1}$ | 0.28 | | −0.01 | | −0.31 | |
| **Bray1-P** | mg P kg$^{-1}$ | 0.20 | | 0.14 | | −0.28 | |
| **Bray2-P** | | −0.30 | | 0.47 | * | 0.14 | |
| **Exchangeable Al** | cmol$_c$ kg$^{-1}$ | −0.48 | ** | 0.13 | | 0.49 | ** |
| **Exchangeable Na** | | −0.73 | ** | 0.28 | | 0.71 | ** |
| **Exchangeable K** | | −0.56 | ** | 0.22 | | 0.54 | ** |
| **Exchangeable Ca** | | −0.60 | ** | 0.10 | | 0.64 | ** |
| **Exchangeable Mg** | | −0.44 | * | −0.03 | | 0.51 | ** |
| **CEC** | | −0.82 | ** | 0.13 | | 0.87 | ** |
| **BS** | % | 0.09 | | −0.02 | | −0.10 | |
| **EBS** | | −0.07 | | 0.04 | | 0.06 | |

C, carbon; N, nitrogen; KCl, potassium chloride; EC, electrical conductivity; BS: base saturation; EBS: effective base saturation; CEC: cation exchange capacity

* $p < 0.05$

** $p < 0.01$

**Table 3. Soil water conditions and fertilization effects based on field experiments from 2013–2015.**

| Site | Soil type | Duration of submergence (Days) | Cumulative water depth (cm) | Grain yield (Mg·ha$^{-1}$) | | | | Increase in grain yield (Mg·ha$^{-1}$) | | |
|---|---|---|---|---|---|---|---|---|---|---|
| | | | | CT | NK | NK+PR | NK+TSP | NK | NK+PR | NK+TSP |
| **Nassoulou** | Luvisols | 29 ± 2 | 114 ± 16 | 0.35 ± 0.18 | 0.93 ± 0.61 | 4.03 ± 0.86 | 4.82 ± 1.02 | 0.58 ± 0.46 | 3.68 ± 0.85 | 4.47 ± 0.99 |
| **Siguinvouse** | Lixisols | 26 ± 1 | 105 ± 4 | 0.67 ± 0.34 | 1.62 ± 0.47 | 2.04 ± 0.88 | 4.28 ± 1.75 | 0.95 ± 0.62 | 1.37 ± 1.22 | 3.61 ± 1.49 |
| **Ramongo** | Lixisols | 23 ± 3 | 90 ± 18 | 0.34 ± 0.05 | 1.52 ± 1.69 | 3.78 ± 1.43 | 3.52 ± 0.21 | 1.19 ± 1.67 | 3.45 ± 1.47 | 3.18 ± 0.25 |
| **Poa** | Lixisols | 40 ± 6 | 229 ± 95 | 2.10 ± 0.26 | 2.45 ± 0.28 | 3.61 ± 0.55 | 3.76 ± 1.51 | 0.35 ± 0.54 | 1.51 ± 0.68 | 1.67 ± 1.28 |
| **Nandiala** | Cambisols | 84 ± 25 | 815 ± 292 | 1.28 ± 0.84 | 1.48 ± 0.46 | 2.99 ± 1.38 | 3.01 ± 1.11 | 0.20 ± 1.11 | 1.71 ± 2.11 | 1.73 ± 0.42 |
| **Villy** | Cambisols | 53 ± 3 | 380 ± 208 | 2.09 ± 0.21 | 2.78 ± 0.70 | 3.53 ± 0.92 | 2.91 ± 1.77 | 0.69 ± 0.49 | 1.44 ± 0.71 | 0.82 ± 1.56 |
| **Sissene** | Cambisols | 45 ± 12 | 322 ± 224 | 2.10 ± 0.28 | 3.04 ± 1.12 | 4.62 ± 1.49 | 6.33 ± 3.79 | 0.93 ± 0.91 | 2.52 ± 1.47 | 4.22 ± 3.53 |
| **Lixisols and Cambisols** | | 29 ± 7 a | 135 ± 71 a | 0.86 ± 0.78 a A | 1.63 ± 0.98 a A | 3.37 ± 1.17 b A | 4.10 ± 1.20 b A | 0.77 ± 0.88 a | 2.51 ± 1.46 b | 3.24 ± 1.42 b |
| **Cambisols** | | 61 ± 23 b | 506 ± 315 b | 1.79 ± 0.64 a B | 2.39 ± 1.03 ab B | 3.74 ± 1.37 ab A | 4.23 ± 2.81 b A | 0.60 ± 0.86 a | 1.95 ± 1.48 a | 2.44 ± 2.51 a |
| **All sites** | | 43 ± 22 | 294 ± 279 | 1.24 ± 0.85 a | 1.93 ± 1.05 a | 3.52 ± 1.23 b | 4.15 ± 1.94 b | 0.70 ± 0.86 a | 2.28 ± 1.45 b | 2.91 ± 1.91 b |

Values represent the mean ± standard deviation (SD) for the three years. Different lowercase letters indicate significant differences between the fertilizer treatments in each soil type. Different uppercase letters indicate significant differences between the soil types. In 2015, no P fertilizer was applied. CT, without fertilization; NK, without P fertilization with N and K fertilization; NK+PR, PR application with N and K fertilization; NK+TSP, TSP application with N and K fertilization; N, nitrogen; P, phosphorus; PR, phosphate rock; K, potassium.

submergence durations and greater cumulative water depths than Lixisols and Luvisols. In Lixisols and Luvisols, the application of NK did not significantly increase grain yield. However, the NK+PR and NK+TSP treatments resulted in significantly higher grain yields than those in the CT and NK treatments. In Cambisols, the only significant difference in grain yield was observed between the CT and NK+TSP treatments. The mean grain yield across all years and sites ranged from $1.24 \pm 0.85$ Mg ha$^{-1}$ for the CT treatment to $4.15 \pm 1.94$ Mg ha$^{-1}$ for the NK+TSP treatment. The application of NPK (NK+TSP and NK+PR) significantly increased grain yield, whereas NK application without P did not. Generally, grain yield was higher in Cambisols than in Lixisols and Luvisols, and a significant difference in grain yield between soil types was only observed in the CT and NK treatments. Furthermore, the increase in grain yield over that in the control treatment (CT) was significantly higher in the NK+TSP and NK+PR treatments compared with that in the NK treatment, although no significant differences were observed between the treatments in Cambisols.

## Geographical characteristics of the seven observation points and inundation score estimation

The water accumulation scores and convexities are summarized in Table 4 (additional topographic features are shown in S2 Fig). Water accumulation scores generally corresponded with the positions of the sites on the slope and elevation, as shown in Fig 1. Nassoulou and Siguinvouse exhibited small water accumulation and positive longitudinal convexity. In contrast, the other five sites exhibited water accumulation values >0.75, with varying longitudinal and cross-sectional convexities. Longitudinal convexity generally matched water accumulation, whereas cross-sectional convexity did not (Fig 3).

Furthermore, water accumulation scores and longitudinal convexity explained the distribution of the soil types and soil water conditions. Specifically, Cambisols were found on sites with water accumulation scores >0.76 and longitudinal convexity <−0.0052 (Fig 3A and Table 4). The duration of submergence and cumulative water depth were longer and deeper on sites with large water accumulation scores and large negative longitudinal convexity (Fig 3B).

The PCA demonstrated the characteristics of the different soil types and soil water conditions (Table 5 and Fig 4). Notably, PC1 accounted for 54% of the total data variance, whereas PC2–PC4 explained 10–16%. Variation between the two soil groups was explained by PC1, whereas variation within the same soil group was explained by PC2.

**Table 4. Geographical characteristics and the estimated inundation score of the seven observation points.**

| Site | Soil type | Water accumulation score | Convexity | | Inundation score |
|---|---|---|---|---|---|
| | | | Longitudinal | Cross-sectional | |
| **Nassoulou** | Luvisols | 0.48 | 5.91E−03 | −2.20E−05 | 0.00 |
| **Siguinvouse** | Lixisols | 0.63 | 1.69E−04 | −8.23E−03 | 0.00 |
| **Ramongo** | Lixisols | 0.80 | −3.91E−03 | −2.93E−03 | 0.31 |
| **Poa** | Lixisols | 0.82 | −4.96E−03 | −1.02E−02 | 0.41 |
| **Nandiala** | Cambisols | 0.76 | −5.23E−03 | −6.31E−03 | 0.40 |
| **Villy** | Cambisols | 0.86 | −1.23E−02 | −1.96E−03 | 1.06 |
| **Sissene** | Cambisols | 0.89 | −8.28E−03 | −8.99E−03 | 0.74 |
| **Lixisols and Luvisols** | | 0.68 | −6.98E−04 | −5.35E−03 | 0.18 |
| **Cambisols** | | 0.84 | −8.60E-03 | −5.75E−03 | 0.73 |

Water accumulation was obtained using the spatial analysis tool in the ArcGIS® software v. 10.7.1. Longitudinal and cross-sectional convexities were generated using ENVI® v. 5.5.1.

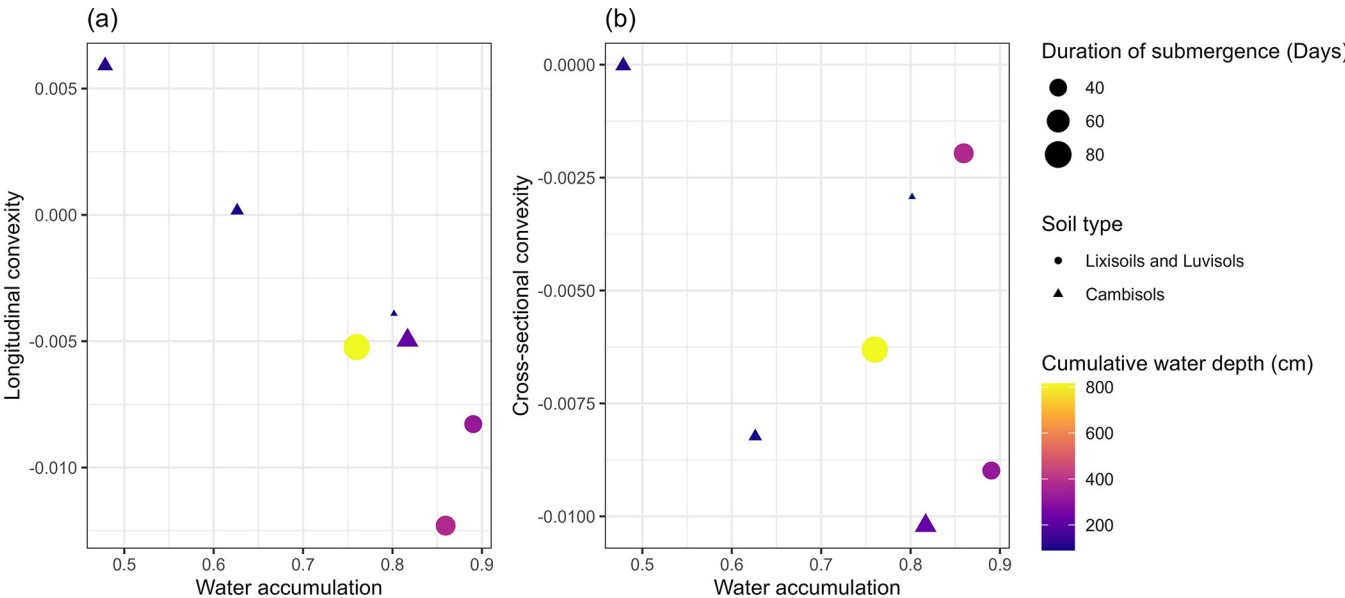

**Fig 3.** Relationships between (A) water accumulation and longitudinal convexity and (B); water accumulation and cross-sectional convexity. Lixisols, Luvisols, and Cambisols: Soil type in the World Reference Base for Soil Resources (WRB) system [30].

Furthermore, the water accumulation scores and longitudinal convexities strongly contributed to PC1. Based on these findings, we used the water accumulation scores and longitudinal convexities to estimate the inundation score. Cambisols had larger inundation scores (0.73 on average) than Lixisols and Luvisols (0.18 on average) (Table 4).

## Integration of field observations and remote sensing data

The inundation score exhibited a significant negative correlation with sand content but a positive correlation with silt and clay content ($p < 0.05$) (Fig 5). If Nandiala, which exhibited an exceptionally long duration of submergence and deep cumulative water depth, was excluded,

**Table 5. Results of the principal component analysis.**

| Importance of components | | PC1 | PC2 | PC3 | PC4 |
|---|---|---|---|---|---|
| Standard deviation | | 2.20 | 1.22 | 1.06 | 0.94 |
| Proportion of variance | | 0.54 | 0.16 | 0.13 | 0.10 |
| Cumulative Proportion | | 0.54 | 0.70 | 0.83 | 0.93 |
| Factor loading values of variables | | PC1 | PC2 | PC3 | PC4 |
| Cumulative precipitation | mm | 0.02 | 0.17 | −0.72 | −0.63 |
| Water accumulation score | | 0.33 | −0.47 | −0.02 | −0.17 |
| Longitudinal convexity | | −0.39 | 0.29 | −0.10 | 0.25 |
| Cross-sectional convexity | | −0.11 | 0.57 | 0.43 | −0.44 |
| Duration of submergence | days | 0.34 | 0.31 | −0.17 | 0.45 |
| Cumulative water depth | cm | 0.32 | 0.34 | −0.39 | 0.29 |
| Sand content | % | −0.42 | −0.16 | −0.20 | 0.11 |
| Silt content | % | 0.41 | −0.16 | 0.09 | −0.09 |
| Clay content | % | 0.40 | 0.28 | 0.23 | −0.12 |

PC, Principal component.

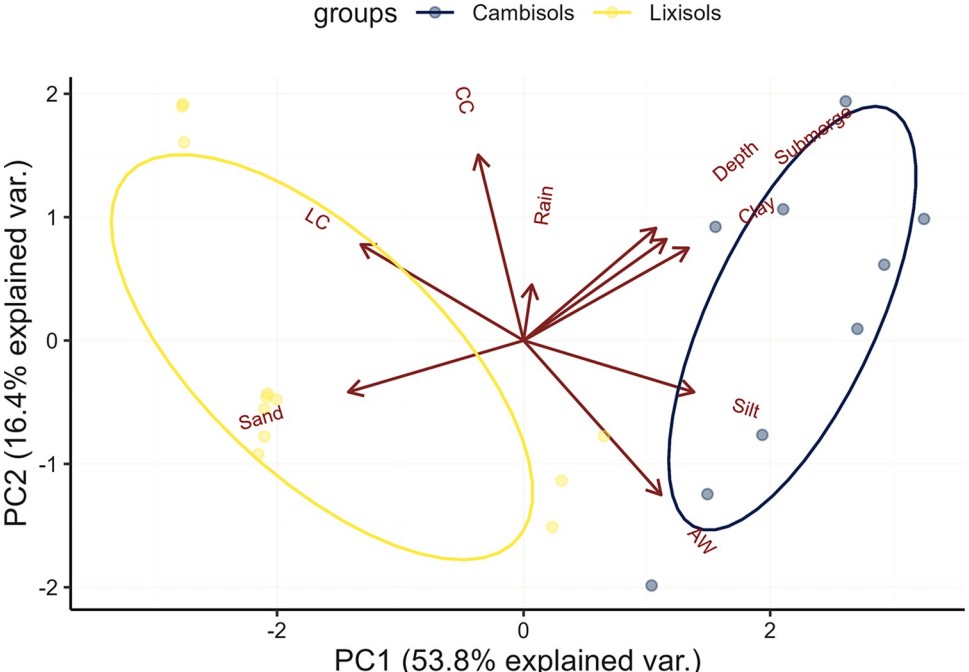

**Fig 4. Biplot of principal component analysis.** PC, principal component; Rain, cumulative precipitation during the growing season; WA, water accumulation score; LC, longitudinal convexity; CC, cross-sectional convexity; Submerge, submergence duration; Depth, cumulative water depth.

the inundation score accounted for 76 and 84% of the total variation in the duration of submergence and cumulative water depth, respectively (Fig 5). There were significant positive relationships between the inundation score and grain yield in the CT and NK treatments, with almost similar slopes in the regression equation (Fig 6A). However, this relationship was not observed in the NK+PR and NK+TSP treatments. Moreover, there was a significant positive correlation between the inundation scores and relative yields of the NK+PR treatment over the NK+TSP treatment (Fig 6B).

## Discussion

### Soil type distribution and the estimation of soil water conditions using inundation scores

The seven soil profiles were classified as Lixisols, Luvisols, and Cambisols (Table 1). The soil horizons were predominantly classified based on the presence of gleyic properties caused by groundwater movement and the presence of an argic horizon. Gleyic properties, without reducing conditions, were observed in Siguinvouse, Nassoulou, and Poa, indicating temporal groundwater impact. In the study area, the GWL gradually rises at the start of the rainy season, saturates with water up to the soil surface level during the rainy season, and sinks to $\geq 100$ cm below the soil surface during the dry season [13]. Hence, both groundwater and stagnant water impact the soil profile morphology and create an albic horizon in soils.

The main differentiator among the soil groups was the presence of an argic horizon starting within 100 cm below the soil surface. Soils with argic horizons, e.g., Lixisols and Luvisols, are generated on stable landforms and have an advanced age of pedogenesis [36]. Topography is important in determining soil distribution in SSA [18, 20, 37]. Ikazaki et al. [18] showed that the soil type in this region changes even within a 12 m elevation. Hence, the local soil types in

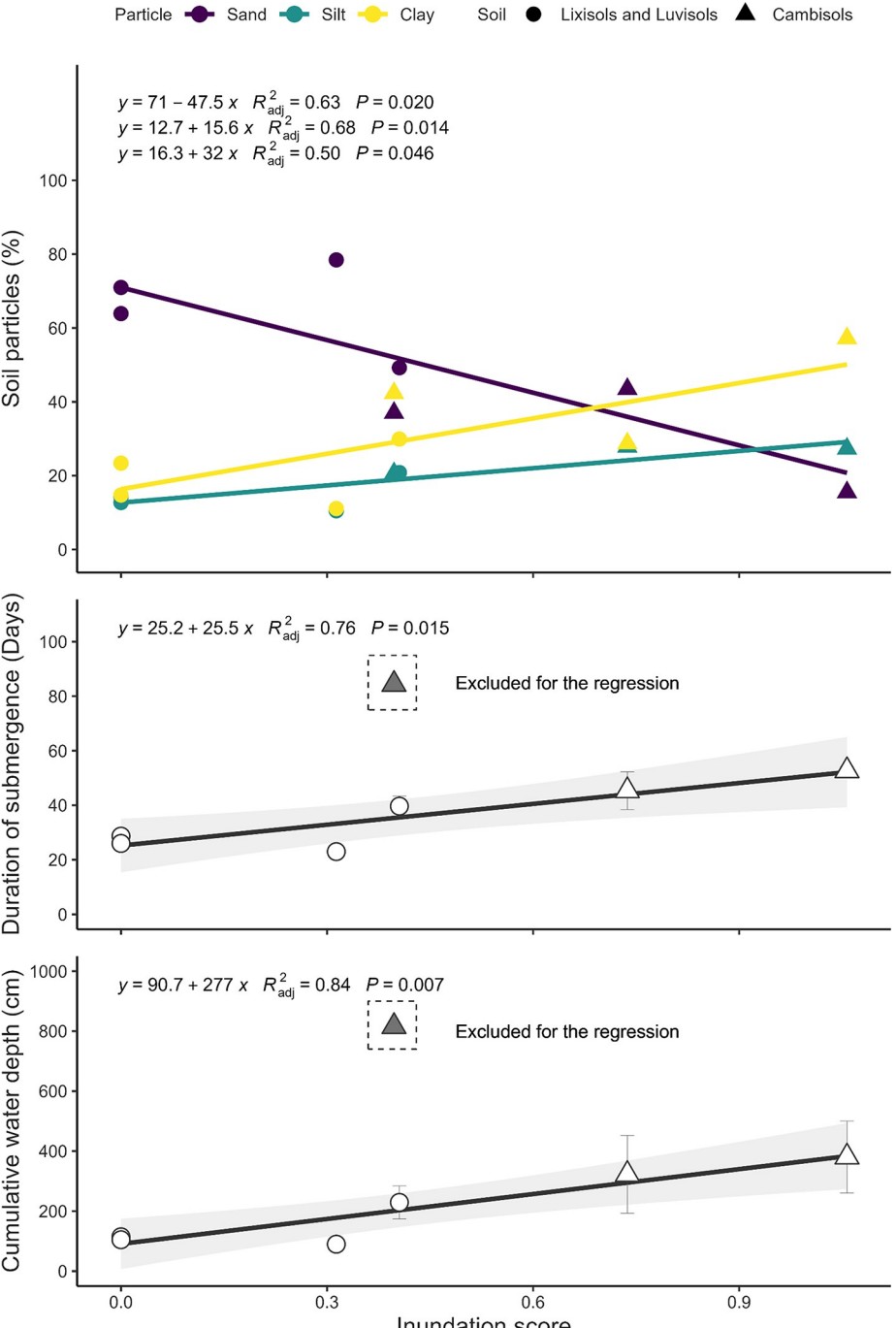

**Fig 5. Relationships between inundation score and soil particle distribution, submergence duration, and cumulative water depth.** Points represent the mean ± standard error for the three years (2013–2015). Soil particles represent the weighted average sand, silt, and clay content in the 0–20 cm soil layer. If the relationship was significant, regression lines and equations were drawn ($p < 0.05$). $R^2_{adj}$, adjusted coefficient of determination; gray region, 95% confidence interval.

this area differ, even on a small scale. Here, Cambisols were located at relatively low altitudes (an average of 296.7 m), whereas Lixisols and Luvisols were located at comparatively higher elevations (an average of 312.8 m).

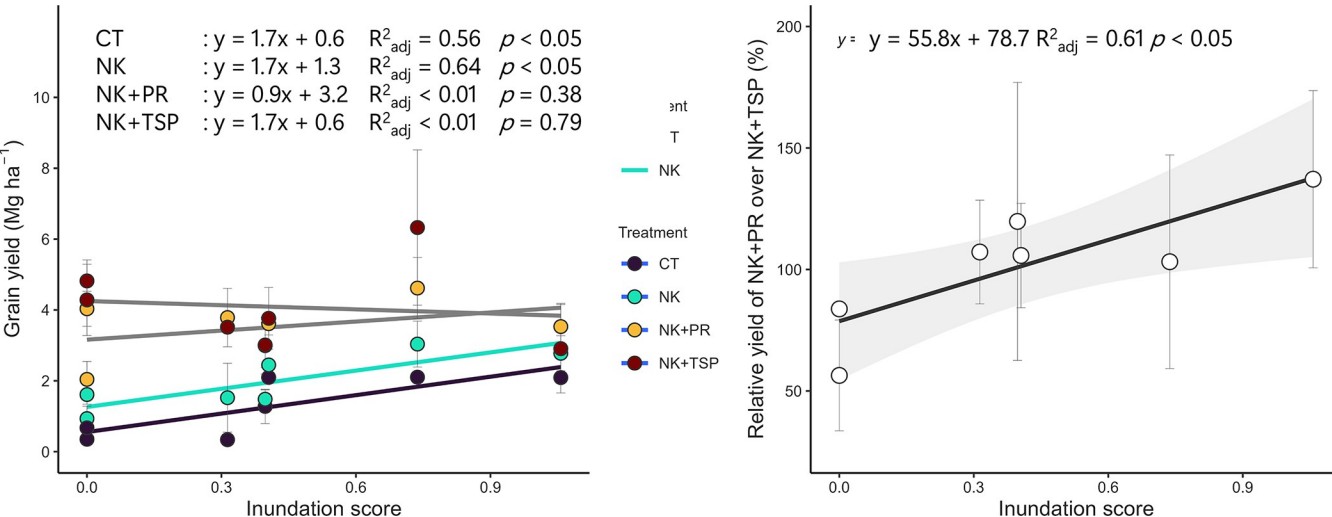

**Fig 6.** Relationships between inundation score and rice grain yield in the different fertilization treatments (A) and between the relative grain yield of PR+NK over TSP+NK (B). Points represent the mean ± standard error for the three years (2013–2015). If the relationship was significant, regression lines and equations were drawn ($p < 0.05$). CT, without fertilization; NK, without P fertilization with N and K fertilization; NK+PR, PR application with N and K fertilization; NK+TSP, TSP application with N and K fertilization; N, nitrogen; P, phosphorus; PR, phosphate rock; K, potassium; $R^2_{adj}$, adjusted coefficient of determination; gray region, 95% confidence interval.

The main physicochemical soil property distinguishing the Lixisols, Luvisols, and Cambisols was the surface horizon soil texture (Table 1). The silt and clay content of the surface horizons of Lixisols and Luvisols (Ramongo, Siguinvouse, Poa, and Nassoulou) were low to medium (silt: 10.8–25.4%; clay: 6.9–19.6%). In contrast, the silt and clay content were high for the three Cambisol profiles (silt: 20.6–30.3%; clay: 30.5–48.3%). The high silt and clay content in Cambisols is attributed to soil erosion and silt and clay illuviation from the upper and middle slopes to the lower slope and river bottom [20]. In Burkina Faso, intense rainfall occurs during the rainy season, causing surface water runoff without underground penetration. Wakatsuki et al. [38] reported that surface runoff deposits clay particles in the surface soil of managed rice fields in Burkina Faso. Ikazaki et al. [18] stated that the effective soil depth thickness for crop production increases in the lower part of the slope. In contrast, the clay in the argic horizon in the Lixisols and Luvisols is attributed to illuvial accumulation from the overlying horizon [18, 20]. Deressa et al. [20] reported that in western Ethiopia, the argic horizon depths in the profiles correlate with the wetting front depths and subsequent drying.

In this study, we utilized DEM and GIS to estimate the inundation scores and explain the distribution of the soil types and soil water conditions (Fig 5). It was found that Cambisols were only present in areas with an inundation score >0.40. The inundation score is determined by the water accumulation score and longitudinal convexity, key factors that affect soil erosion and clay illuviation [39, 40]. Therefore, the inundation score can be utilized in predicting the distribution of soil types on a watershed scale. The significant correlation between the inundation score and soil particle distribution emphasizes the illuvial accumulation of fine soil particles with water movement (Fig 6A).

Notably, Lixisols were also found in areas with an inundation score of 0.41. This can be partly explained by the fact that Lixisols have a wider range of soil characteristics compared to Cambisols, as demonstrated by PC1 (Fig 4). For example, soil in Poa was classified as Lixisols. However, the distribution of soil particles in the profiles showed a different pattern (Table 1). Although the argic horizon was observed in the soil profile of Poa, the silt and clay content in the surface horizon was relatively higher compared with that at the other Lixisol and Luvisol

sites. Additionally, the argic horizon in Poa was found at a depth of 9.0 cm from the surface. These findings suggest that the soil in Poa is situated in the transition zone between Lixisols and Cambisols. It is essential to conduct surveys at multiple locations covering the watershed in future studies to normalize the results.

The inundation score explained the field observations of submergence duration and cumulative water depth, indicating that the inundation score can be utilized for further analysis to understand P fertilization efficacy (Fig 6). However, we needed to exclude Nandiala, with its extremely long submergence duration and deep cumulative water depth, which may be attributed to the impact of groundwater fractionation or inundation from natural water basins, important factors determining water conditions in this region [12, 13]. Additionally, based on the field observations, the site is located near a road, which obstructs the flow of water and has an impact on the submergence conditions.

## Inundation score as an indicator of the fertilization effect

In this study, the grain yield for the CT and NK+TSP treatments may reflect the original soil productivity and the potential yield of rice cultivation, respectively. The grain yield in the NK treatment (average $1.93 \pm 1.05$ Mg ha$^{-1}$) was significantly lower than that in the NK+TSP treatment (average $4.15 \pm 1.94$ Mg ha$^{-1}$) (Table 3). Considering the proposed critical limit of the Bray1-P content based on previous studies in SSA, which is 7–9 mg P kg$^{-1}$ [41], the Bray1-P content in this study were substantially low, ranging from 0.9–6.1 mg P kg$^{-1}$ (Table 1). These results indicate that P deficiency was a critical limiting factor for rice productivity, as reported in SSA [6, 42]. However, grain yield in NK showed an increasing trend compared with CT. In addition, recent studies in SSA showed that cases of P and K deficiency alone did not frequently occur during rainfed lowland rice cultivation; however, their interactions with N deficiency are an important limiting factor in rice production [6, 17]. Furthermore, Dossou-Yovo et al. [43], compiled 652 field experiments in SSA and showed that the soil total N content was not invariably related to soil N mineralization and rice yield without N application. Based on these results, P is the critical yield-limiting factor in rainfed lowland rice cultivation in this region, along with an overall nutrient deficiency. Notably, $3.52 \pm 1.23$ Mg ha$^{-1}$ of grain yield in the NK+PR treatment was far larger than the 2.6 Mg ha$^{-1}$ of the average yield of rainfed lowland rice cultivation in SSA [2], indicating the potential utilization of low-grade PR to increase rice production in SSA. However, there are conflicting data regarding the effect of the direct application of PR on lowland rice cultivation in SSA. According to previous studies, soil water conditions are critical factors in controlling the effect of PR and PR-based local P fertilizers effect [12–14]. Although soil type is often considered an important factor that impacts the efficacy of fertilization, in this study, the soil water conditions and soil type distribution were closely related and explained based on inundation scores, as discussed previously. Therefore, these two factors can be addressed together.

Significant positive relationships were observed between the inundation scores and grain yield for the CT and NK treatments but not for the treatment with P fertilization (Fig 6A). Furthermore, the regression slopes for the CT and NK treatments were almost parallel, with values of 1.73 and 1.71 for the CT and NK treatments, respectively, and the $y$-intercepts differed at 0.56 and 1.26 Mg ha$^{-1}$ for CT and NK, respectively. These relationship trends suggest that N and K limit grain yield, regardless of the differences in inundation scores. Therefore, grain yield increased after NK application in Lixisols and Luvisols, and Cambisols showed similar values. The significant positive relationships between the inundation scores and grain yield for the CT and NK treatments can be explained by soil particle distribution and chemical properties (Table 2). Clay and silt particles contribute to the physical and chemical stabilization of

soil organic matter, which affects soil C and N accumulation [44, 45], and clay particles exhibit a high CEC [46]. The findings of previous studies support significant positive correlations between silt or clay content and total C and N content, exchangeable cations, and CEC, with high correlation coefficients. Moreover, water erosion can transport macro- and micronutrient elements, as reported globally, including in Africa [47, 48]. Although there was no significant correlation between soil particles and Bray1-P content, Bray2-P content positively correlated with silt content (Table 2). Thus, the illuvial accumulation of silt particles and the transportation of P at sites with high inundation scores resulted in high grain yields in the CT and NK treatments. Moreover, yield limitation due to low Bray2-P content was relatively more extensive in soils with low inundation scores, resulting in a drastic increase in grain yields after P application in Lixisols and Luvisols.

Finally, we found that the relative grain yields of the NK+PR treatment over the NK+TSP treatment increased at sites with high inundation scores (Fig 6B). Iwasaki et al. [13] revealed that rice plants in the central plateau of Burkina Faso could utilize the relatively soluble P fraction in fields with high GWLs. In this study region, the rice cultivation area is characterized by alternative inundation and drying, even in the lowland (corresponding to the sites with high inundation scores in this study), particularly in the early growing stage [13]. These results were probably due to the positive effects of alternative drying and rewetting on the acceleration of root architecture development and P diffusion in the soil [49] at the sites with high inundation scores.

Furthermore, there was a large variation in yields under P fertilization conditions at similar flooding scores. Thus, concerns remain regarding the relatively small area under evaluation in this study. Therefore, it is desirable to increase the number of study sites in future studies and validate the results obtained in this study.

## Conclusions

Inundation scores can be used to estimate the distribution of soil types in lowland areas along rivers and the effects of different sources of P fertilization while considering the interactions between soil water conditions, the accumulation of soil particles through illuviation, and the movement of nutrients. Regardless of the soil type, rice grain yields were significantly limited by P deficiency. However, the effect of P fertilization and the optimal selection of P fertilizer types differed based on the inundation score. At sites with low inundation scores (mainly Lixisols and Luvisols), a drastic increase in grain yields can be expected after TSP application. Conversely, at sites with high inundation scores, although the P fertilization effect was relatively small, the PR application showed a performance comparable to that of the TSP application. These results provide basic information for determining optimal P fertilization in the upper reaches of rivers in Burkina Faso and SSA. However, further soil profiling surveys and observations for soil water conditions are essential to bridge the gap between remote sensing data and field-scale observations.

## Supporting information

**S1 Fig. Topographical characteristics of the study sites.**
(DOCX)

**S2 Fig.** Photographs of the soil profiles (upper panel) and soil texture distributions (lower panel). Horizon names were set following the World Reference Base for Soil Resources [30].
(DOCX)

**S1 Table. Soil morphological properties based on the World Reference Base for Soil Resources [30].** Mottling is described in the order of abundance, contrast, shape, and species. The mottled color is expressed in parentheses. Abbreviations: N, None; V, Very few (0–2%); F, few (2–5%); C, Common (5–15%); M, Many (15–40%); A, Abundant (>40%). Contrast: F, Faint; D, Distinct; P, Prominent. Shape: RO, Root-like; FI, Filmy; IR, Irregular; SP, Speckled; CL, Cloudy. Species: Fe, Iron; Mn, Manganese. Concretions are described based on abundance and species. The abbreviations are the same as those used for mottling. The concentrations of $Fe2+$ ions were related to the strength of the reaction with the 0.2% $\alpha$-dipyridyl solution in 10% acetic acid. Strength of reaction:–, none; ±, very weak; +, weak; ++, strong; +++, very strong. Rocks are described in the order of abundance, weathering state, size, and shape. Abbreviations: N, None; F, Few (0–5%); C, Common (5–10%); M, Many (10–20%); A, Abundant (20–50%); D, Dominant (>50%). State of weathering: F, Fresh; SL, Slightly weathered; W, Weathered; ST, Strongly weathered. Size: FC, Fine gravel (0.2–1 cm); G, Gravel (1–5 cm); S, Stone (5–10 cm); LS, Large stone (10–20 cm); UN Unknown. The structure of the rocks is described in the order of their grades, sizes, and types. Grade: W, Weak; M, Moderate; S, Strong. Size: VF, Very fine (<5 mm); F, Fine (5–10 mm); M, Medium (10–20 mm); C, Coarse (20–50 mm); VC, Very coarse (>50 mm). Type: SB, Subangular block; PL, Platy; M, Massive; N, None.
(DOCX)

## Acknowledgments

The authors thank the local farmers and technical staff members for their support with field trials and the Soil Team at JIRCAS for soil chemical analysis.

## Author Contributions

**Conceptualization:** Takashi Kanda, Satoshi Nakamura, Albert Barro, Fujio Nagumo.

**Data curation:** Shinya Iwasaki, Takashi Kanda, Simporé Saïdou.

**Formal analysis:** Shinya Iwasaki, Takashi Kanda, Simporé Saïdou.

**Funding acquisition:** Satoshi Nakamura, Fujio Nagumo.

**Investigation:** Shinya Iwasaki, Satoshi Uchida.

**Methodology:** Shinya Iwasaki, Satoshi Nakamura, Satoshi Uchida.

**Project administration:** Albert Barro, Fujio Nagumo.

**Resources:** Simporé Saïdou.

**Supervision:** Satoshi Nakamura, Albert Barro.

**Visualization:** Shinya Iwasaki.

**Writing – original draft:** Shinya Iwasaki.

**Writing – review & editing:** Takashi Kanda, Satoshi Nakamura, Satoshi Uchida, Simporé Saïdou, Fujio Nagumo.

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
