## [Decision Letter · Decision Letter 0]

28 Feb 2024

PONE-D-23-41713Integrating Field Surveys and Remote Sensing to Optimize Phosphorus Resource Management for Rainfed Rice Production in the Central Plateau of Burkina FasoPLOS ONE

Dear Dr. Iwasaki,

Thank you for submitting your manuscript to PLOS ONE. After careful consideration, we feel that it has merit but does not fully meet PLOS ONE’s publication criteria as it currently stands. Therefore, we invite you to submit a revised version of the manuscript that addresses the points raised during the review process. **Please follow the comments provided by the reviewer to prepare a revised version of your manuscript.**

We look forward to receiving your revised manuscript.

Kind regards,

Paulo H. Pagliari

Academic Editor

PLOS ONE

Journal Requirements:

This work was financially supported by the Science and Technology Research Partnership for Sustainable Development (SATREPS) project No. JPMJSA1609, Japan Science and Technology Agency (JST), and Japan International Cooperation Agency (JICA) (Project on the establishment of the model for fertilizing cultivation promotion using Burkina Faso phosphate rock, No. JPMJSA1609).

4. In the online submission form, you indicated that Data is available upon request.

Reviewers' comments:

Reviewer's Responses to Questions

**Comments to the Author**

1. Is the manuscript technically sound, and do the data support the conclusions?

Reviewer #1: Yes

2. Has the statistical analysis been performed appropriately and rigorously? 

Reviewer #1: Yes

3. Have the authors made all data underlying the findings in their manuscript fully available?

Reviewer #1: Yes

4. Is the manuscript presented in an intelligible fashion and written in standard English?

Reviewer #1: Yes

5. Review Comments to the Author

Reviewer #1: According to the authors, the study aimed to identify the soil type distribution and factors impacting the P fertilization effect on lowland rice production by integrating a field survey and an estimation of soil water conditions using a remote sensing approach.

- The topic is very important as in most of the soils of sub-Saharan Africa, crop production is limited by the low P availability of soils. Moreover, knowing the factors influencing phosphate fertilizers efficiency will allow a site-specific recommendation of fertilizer types and amount and then maximize crop production.

However, the objective of the work must be reformulated to show the relevance of the study. In fact, through the manuscript, the work is about to evaluate the water condition, mainly inundation score of rice cultivation area and its useability in recommending phosphorus fertilizers for rice production. Moreover, previous studies in the studied area have highlighted the relationship between soil chemical and physical characteristics and the response of crops to phosphorus fertilizers (Nakamura et al., 2020, Fukuda et al., 2021, Iwasaki et al., 2021; Soma et al., 2023).

-Line 122 Materials and methods: define the expression “local communities” and explain why you selected farmers by “local communities”.

- Materials and methods: Is one farmer per village for a total of 7 villages representative of the studied area? How the authors expect to manage phosphorus fertilization using inundation score in some area such as Nandiala, as some time this location was drop during data analysis?

- Results: There was not a significant relationship between inundation score and yield in soil receiving P fertilizers. Can we still use this parameter to map P fertilization? More explanations needed.

6. PLOS authors have the option to publish the peer review history of their article (what does this mean?). If published, this will include your full peer review and any attached files.

Reviewer #1: No

---

## [Author Response · Author response to Decision Letter 0]

9 May 2024

To Reviewer 1

Thank you very much for your constructive and accurate comments on our manuscript.

We have revised the manuscript according to your comments.

The major changes made are outlined below:

1. Style Revision: We have modified the style throughout the manuscript to align with the journal's guidelines.

2. Selection Method and Research Permissions: We have revised the sections concerning the selection method of the research site and obtaining permissions for the research to ensure clarity and compliance.

3. Limitations and Future Prospects: We have included a section discussing the limitations of our study and outlining potential avenues for future research.

Reviewer #1: 

C: According to the authors, the study aimed to identify the soil type distribution and factors impacting the P fertilization effect on lowland rice production by integrating a field survey and an estimation of soil water conditions using a remote sensing approach. The topic is very important as in most of the soils of sub-Saharan Africa, crop production is limited by the low P availability of soils. Moreover, knowing the factors influencing phosphate fertilizers efficiency will allow a site-specific recommendation of fertilizer types and amount and then maximize crop production.

However, the objective of the work must be reformulated to show the relevance of the study. In fact, through the manuscript, the work is about to evaluate the water condition, mainly inundation score of rice cultivation area and its useability in recommending phosphorus fertilizers for rice production. Moreover, previous studies in the studied area have highlighted the relationship between soil chemical and physical characteristics and the response of crops to phosphorus fertilizers (Nakamura et al., 2020, Fukuda et al., 2021, Iwasaki et al., 2021; Soma et al., 2023).

A: Thank you for pointing this out. I think you are exactly right.

We have revised the following (L101-104)

Therefore, this study aims to evaluate the water condition of rice cultivation area and its useability in recommending P fertilizers for rice production. Achieving these objectives could contribute to increasing rice productivity in the upper reaches of rivers in Burkina Faso and SSA

C:-Line 122 Materials and methods: define the expression “local communities” and explain why you selected farmers by “local communities”.

- Materials and methods: Is one farmer per village for a total of 7 villages representative of the studied area? How the authors expect to manage phosphorus fertilization using inundation score in some area such as Nandiala, as some time this location was drop during data analysis?

C: - Results: There was not a significant relationship between inundation score and yield in soil receiving P fertilizers. Can we still use this parameter to map P fertilization? More explanations needed.

A： Thank you for bringing this to our attention. We have incorporated the following revisions in response to the editor's comments.

During the site selection process, it was imperative to choose plots with varying topography and water environments, spanning from upstream to downstream of the watershed. Subsequently, we obtained survey permissions from the farmers through INERA. While at one site (Nandiala), where man-made impacts were evident, the inundation score estimated from the GIS proved inadequate, at other sites, it was estimated with a high degree of accuracy. These findings highlight the limitations of our study and underscore areas for future improvement.

Please see below:

L107-113

L476-481

---

## [Decision Letter · Decision Letter 1]

2 Jul 2024

PONE-D-23-41713R1Integrating Field Surveys and Remote Sensing to Optimize Phosphorus Resource Management for Rainfed Rice Production in the Central Plateau of Burkina FasoPLOS ONE

Dear Dr. Iwasaki,

Thank you for submitting your manuscript to PLOS ONE. After careful consideration, we feel that it has merit but does not fully meet PLOS ONE’s publication criteria as it currently stands. Therefore, we invite you to submit a revised version of the manuscript that addresses the points raised during the review process.

We look forward to receiving your revised manuscript.

Kind regards,

Paulo H. Pagliari

Academic Editor

PLOS ONE

Additional Editor Comments:

More revisions are needed. Please provided a copy with the detailed changes so that the reviewers can assess the changes made. thank you

Reviewers' comments:

Reviewer's Responses to Questions

**Comments to the Author**

1. If the authors have adequately addressed your comments raised in a previous round of review and you feel that this manuscript is now acceptable for publication, you may indicate that here to bypass the “Comments to the Author” section, enter your conflict of interest statement in the “Confidential to Editor” section, and submit your "Accept" recommendation.

Reviewer #2: (No Response)

2. Is the manuscript technically sound, and do the data support the conclusions?

Reviewer #2: Yes

3. Has the statistical analysis been performed appropriately and rigorously? 

Reviewer #2: Yes

4. Have the authors made all data underlying the findings in their manuscript fully available?

Reviewer #2: No

5. Is the manuscript presented in an intelligible fashion and written in standard English?

Reviewer #2: Yes

6. Review Comments to the Author

Reviewer #2: Review report

1. Summary of the research and your overall impression

The manuscript aims to investigate the factors influencing the efficiency of rock phosphate application on rainfed lowland rice production in Burkina Faso and to generalize the findings using remote sensing data. The topic addressed in this article is worthy of investigation, given the importance of finding alternatives to the costly mineral fertilizers currently in use. I also acknowledge the efforts invested in identifying the soil type in the seven fields. However, there are some questionable conclusions, flaws in the writing, and missing information that I believe should be addressed to improve the scientific quality of the paper. Therefore, I recommend revising the paper by providing a more coherent interpretation and including the missing information. While I apologize for providing so many remarks at this stage of the review process, please remember that my sole objective is to improve your paper.

2. Specific areas for improvement

2.1. Major issues

• Materials and methods

o Lines 150-152 Why did you cite Nakamura et al. (2020)?

I do not perceive the link. Are these data already analysed and published? What are the differences with the current dataset?

o Did you check the conditions of application of Pearson’s correlation coefficient? In case the conditions are not met, Spearman’s correlation is an alternative.

o I understand why you combined the data of Luvisols and Lixisols. However, I suggest clearly explaining the reason in the Materials and Methods section.

• Discussion

o To corroborate your claim that P is the most important limiting factor in these sites, I suggest in the discussion to compare the N, P and K values to average values or threshold of soil fertility in rainfed lowland rice fields in sub-Saharan Africa.

2.2. Minor issues

• The pages are not numbered.

• Line 28: Were the duration of submergence and cumulative water level recorded or extracted from remote sensing data?

• Lines 30-32: I do not see the relevance of this paragraph in the Abstract, as these are not key points of this study. Besides, no research objective mentioned in the Abstract is linked to this result. I suggest deleting these sentences

• Lines 33-34: What do you mean by “water environment”? Are you referring to soil water conditions? Please use the appropriate expression.

• Lines 81-82: Is this information on the yield of sorghum relevant for this study?

• Lines 82-84: Is this insight on cowpea relevant to this study? Why not look for studies on rice in West Africa?

• Lines 87-88: Is the French translation necessary? What do you mean by "lowland riverine areas"? This sounds a bit awkward. You could just use "rainfed lowland rice."

• Line 108: I wonder if these are cultivation areas of just villages.

• Line 111: I wonder if these are cultivation areas of just villages.

• Line 113: It seems that the word “pattern” is missing.

• Line 124: Here, you are talking about soil particle size proportion and not texture.

• How did you determine the soil texture? Which textural triangle did you use (see also Table 1)?

• Line 161-162: Please provide more understandable labels for each treatment. Please be clear.

• Line 169: There is redundancy. Please delete “in a planting space”.

• Lines 222: I suggest writing “Table S1” and “Fig. S1”

• Line 229: cmolc. It better to avec “c “as an indice.

• Could you please spell out Bmvg, Btwgc, and Bmv?

• Lines 247-248: Please rephrase. Please avoid interpreting a correlation as a cause-effect relationship.

• Lines272: NK + TSP > NK + PR. This is not statistically different at p < 0.05. Therefore, is writing this true?

• Line 274: Please rephrase the title of Table 3

• Line 286: six or five? 7-2 = 6 or 5?

• Lines 290-292: “Fig 3. Relationships between (a) water accumulation and longitudinal convexity and (b); water accumulation and cross-sectional convexity.

• Lines 301-302: Please rephrase. It does not make sense when expressed like this.

• Line 336: “positive” relationships?

• Line 349: One decimal could be enough.

• Lines 428-431: The focus of this study is rainfed lowland rice, correct? If so, I recommend comparing your average yield with those specific to the rainfed lowland rice production system. The average yield from USDA (2018) includes irrigated lowland, rainfed lowland, and rainfed upland, making the comparison unfair.

• Lines 436: What do you mean by “water environment”? Are you referring to soil water conditions? Please use the appropriate expression.

• Lines 455-458: Why do make this conclusion?

• Line 465: Please rephrase this sentence. Do you mean the relative grain yield of NK + PR over NK +TSP?

• Lines 466-471: The link and interpretation are unclear, making it difficult to grasp the explanation.

• Lines 472: What do you mean by “tentative”?

• Table 1: For the pH, One decimal is enough.

• Table 1: Sand (%), Clay (%)

• Table 2: Relationships between soil particle size proportion and physicochemical properties (n = 34). How did you obtain this sample size of 34?

• Line 476: What is the reference 49?

• Lines 476-477: The sentence “In this study, … analysis” could be deleted. I could not perceive his relevance in this paragraph.

• Line 480-481 “ .. verification results at many more sites can be expected”. What do you mean? Please rephrase this sentence.

• Lines 495: What do you mean by “water environment”? Are you referring to soil water conditions? Please use the appropriate expression.

• Figure 6: Please be consistent with the abbreviation. I would suggest NK + PR and NK + TSP

• Figure 6: Please correct the y-axis of Fig 6b. Do you mean the relative grain yield of NK + PR over NK +TSP?

7. PLOS authors have the option to publish the peer review history of their article (what does this mean?). If published, this will include your full peer review and any attached files.

Reviewer #2: No

---

## [Author Response · Author response to Decision Letter 1]

23 Jul 2024

Reviewers

We greatly appreciate the effort and time reviewers and editors put into improving our manuscript.

We have thoroughly addressed your feedback and comments in this revised version. The major changes made are outlined below:

1. Introduction and discussion: We have modified the introduction and discussion to focus more on rainfed lowland rice cultivation.

2. Style Revision: We have modified the style throughout the manuscript to align with the journal's guidelines. A professional organization edited the manuscript in English.

For further details, please refer to the attached "Response to Reviewers."

If any responses are unclear or you wish to make additional changes, please let us know.

Respectfully yours,

Shinya IWASAKI (Corresponding author)

---

## [Decision Letter · Decision Letter 2]

4 Sep 2024

PONE-D-23-41713R2Integrating Field Surveys and Remote Sensing to Optimize Phosphorus Resource Management for Rainfed Rice Production in the Central Plateau of Burkina FasoPLOS ONE

Dear Dr. Iwasaki,

Thank you for submitting your manuscript to PLOS ONE. After careful consideration, we feel that it has merit but does not fully meet PLOS ONE’s publication criteria as it currently stands. Therefore, we invite you to submit a revised version of the manuscript that addresses the points raised during the review process.

We look forward to receiving your revised manuscript.

Kind regards,

Paulo H. Pagliari

Academic Editor

PLOS ONE

Journal Requirements:

Additional Editor Comments:

Minor changes are needed before we can accept the manuscript for publication

Reviewers' comments:

Reviewer's Responses to Questions

**Comments to the Author**

1. If the authors have adequately addressed your comments raised in a previous round of review and you feel that this manuscript is now acceptable for publication, you may indicate that here to bypass the “Comments to the Author” section, enter your conflict of interest statement in the “Confidential to Editor” section, and submit your "Accept" recommendation.

Reviewer #2: (No Response)

2. Is the manuscript technically sound, and do the data support the conclusions?

Reviewer #2: Yes

3. Has the statistical analysis been performed appropriately and rigorously? 

Reviewer #2: Yes

4. Have the authors made all data underlying the findings in their manuscript fully available?

Reviewer #2: No

5. Is the manuscript presented in an intelligible fashion and written in standard English?

Reviewer #2: Yes

6. Review Comments to the Author

Reviewer #2: Review report

1. Overall impression of the revision

The authors have provided satisfactory responses and carefully revised the manuscript. I am in favor of its publication. However, there are still some minor phrasing issues and inconsistencies. As I was not able to carefully read the entire document again, I strongly suggest that the authors carefully proofread it before the final publication.

All the best,

Here are some examples of phrasing issues:

1.1. Minor issues

• I suggest either specifying which data were already published in Nakamura et al. (2020) or removing this sentence altogether.

• Lines 356-357: Please ensure that "NK+PR" and "NK+TSP" are used consistently throughout the document. Replace any variations or previous abbreviations with these terms.

• Lines 433-435: The phrase should be revised to clarify that it is the phosphorus (P) content in the soil that is low, not the soil itself.

• Lines 443-445: The sentence is difficult to understand. Rephrasing it in a simpler way could be helpful.

• Please use the term "notably" more sparingly.

7. PLOS authors have the option to publish the peer review history of their article (what does this mean?). If published, this will include your full peer review and any attached files.

Reviewer #2: No

---

## [Author Response · Author response to Decision Letter 2]

27 Sep 2024

Reviewers

We greatly appreciate the effort and time reviewers and editors put into improving our manuscript. If any responses are unclear or you wish to make additional changes, please let us know.

Respectfully yours,

Shinya IWASAKI (Corresponding author)

---

## [Editor Report · Decision Letter 3]

30 Sep 2024

Integrating Field Surveys and Remote Sensing to Optimize Phosphorus Resource Management for Rainfed Rice Production in the Central Plateau of Burkina Faso

PONE-D-23-41713R3

Dear Dr. Iwasaki,

We’re pleased to inform you that your manuscript has been judged scientifically suitable for publication and will be formally accepted for publication once it meets all outstanding technical requirements.

Kind regards,

Paulo H. Pagliari

Academic Editor

PLOS ONE
---

## [Editor Report · Acceptance letter]

17 Oct 2024

PONE-D-23-41713R3 

PLOS ONE

Dear Dr. Iwasaki, 

I'm pleased to inform you that your manuscript has been deemed suitable for publication in PLOS ONE. Congratulations! Your manuscript is now being handed over to our production team.

Kind regards, 

on behalf of

Dr. Paulo H. Pagliari 

Academic Editor

PLOS ONE